# A Learning Community Involving Collaborative Course-Based Research Experiences for Foundational Chemistry Laboratories

**David M. Rubush** [1,*] and **Kari L. Stone** [2,*]

1 Department of Chemistry, Benedictine University, Lisle, IL, 60532, USA
2 Department of Chemistry, Lewis University, Romeoville, IL, 60446, USA
* Correspondence: drubush@ben.edu (D.M.R.); kstone1@lewisu.edu (K.L.S.); Tel.: +1-815-834-6109 (K.L.S.)

**Abstract:** Numerous American national committees have recommended the replacement of traditional labs with a more engaging curriculum that inspires inquiry and enhances scientific skills (examples include the President's Council of Advisors on Science and Technology (PCAST)'s Engage to Excel program and American Association for the Advancement of Science (AAAS) Vision and Change, among others), due to a large body of evidence that shows significant enhancements in student learning and affective outcomes. The implementation of Course-Based Undergraduate Research Experiences (CUREs) is a creative way to scale up the deployment of authentic research experiences to students. Another highly regarded high-impact practice in postsecondary education is the addition of learning communities. The integration of a three-course learning community and authentic research experiences to laboratory courses adds both a community of scholarship and a development of scientific communication and process skills. This study describes a course that blends these two high-impact practices in higher education in order to promote greater post-course gains in essential elements of a CURE curriculum. This collaborative course shows large post-course gains in essential elements, such as scientific communication and working collaboratively.

**Keywords:** CURE; learning community; course-based research experiences; retention; foundational chemistry laboratories; STEM laboratory

## 1. Introduction

There is a common need for institutions of higher education to adapt from common modalities of instruction to high-impact pedagogical practices in order to enhance the engagement of college students in the science, technology, engineering, and mathematics (STEM) disciplines [1]. Traditional laboratory instruction in chemistry courses prompts all students in a classroom to complete scripted activities, where they will arrive at an already determined outcome, which fails to inspire the curiosity and creativity that leads research progress among practitioners [2–10]. Exposing students early in their post-secondary education to the processes of science offers the possibility to inspire the next generation of scientists, who are prepared to enter the workforce and to answer the big questions facing this generation.

In a 2012 American presidential report, the President's Council of Advisors on Science and Technology (PCAST) made five key recommendations to address the predicted shortfall of 1 million college graduates in science, technology, engineering, and mathematics (STEM) fields over the next decade [11]. These recommendations aim to increase the persistence of postsecondary students in STEM fields and motivate faculties to engage in evidence-based instructional practices. Two of the PCAST recommendations are to: (1) "catalyze widespread adoption of empirically validated teaching

practices" and (2) "advocate and provide support for replacing standard laboratory courses with discovery-based research courses." Seeking to address the shortfall of educated science and technology graduates, the main strategy is to halt the leaky STEM pipeline by engaging in pedagogical practices that are known to increase retention.

The persistence framework captures the types of activities that may lead to more students entering into STEM fields after college [12]. Embedded authentic research experiences early in post-secondary education and learning communities are key interventions that have shown increased retention of students in STEM disciplines. This concept relies on the observation that motivation is an important criterion for self-efficacy, which is imperative for enhancing engagement in a field of study. The framework is a circular feedback loop, where learning science leads to confidence, which increases motivation, which ultimately leads to a student identifying as a scientist, which, again, leads to confidence and motivation. At the heart of this framework are three evidence-based efforts that leads to these affective gains: (1) early research experiences [6,13], (2) active learning [14–17], and (3) learning communities [18–20]. The persistence framework unifies successful retention models into a few key practices that should be at the center of any STEM curricular reforms.

Numerous national calls to include these programmatic enhancements have lead entire STEM disciplines to creatively deploy curriculum modifications that aim to increase retention and overall experiential learning [1,6,7,11,12,21–28]. Adding authentic research experiences to the laboratory curriculum has the benefit of including all students in the craft of research [8,29–31]. To model the elements of traditional research experiences, the five elements of a Course-Based Undergraduate Research Experience (CURE) curriculum are as follows:

(1) Place a deliberate emphasis on scientific practice in three phases:

   (a) develop a hypothesis, (b) collect and analyze data and (c) communicate findings;

(2) Promote and support the discovery of new knowledge;
(3) Ensure broad relevance to the scientific community;
(4) Incorporate a team-based approach that cultivates scientific collaboration in the laboratory;
(5) Include iterative advances, leading to new questions of discovery, where the answers are not apparent to either the instructor or student.

These elements are similar to those of traditional research experiments, but a course-based approach to authentic research adds the benefit of having a built-in peer group of classmates from the learning community throughout the three courses [5]. Engaging STEM students with authentic research experiences that contain these five elements allows them to gain ownership of their laboratory experience and increases their motivation and self-efficacy through the support of their peers, which is enhanced by the learning community [32–39]. Despite the strong evidence demonstrating the effectiveness of CUREs in laboratory courses, most chemistry laboratory curricula lack the widespread adoption of these experiences, especially in foundational courses designed for first- and second-year undergraduates. Described here is the blending of a collaborative CURE between general chemistry and organic chemistry laboratories, along with a learning community that is tied to a research writing course. The enhancement of these courses by these curricular modifications has led to increased learning and affective gains among the essential elements of a research experience.

## 2. Materials and Methods

### 2.1. Survey Data

The Classroom Undergraduate Research Experience (CURE) survey was administered as a pre-survey before the class had begun and a post-survey, which was held on the last day of class [35–37]. This survey is used to measure student experiences in a research or research-like setting. Students responded to the survey online. Students filled out a waiver form before starting the survey. Student

responses were not anonymous to the study analysts, since the matching of ID numbers was important to link pre- and post-surveys, but raw data was anonymized, so as to remove all identifying information, and each student was given a number for matching.

*2.2. Data Analysis*

These studies were approved by the Institutional Review Board at Benedictine University, confirmation number 20190412B. The first year of the study data analysis was carried out by the Grinnell College team, Dr. David Lopatto and Leslie Jaworski, and the report was sent to the research team and included a comparison with the larger student data set. In the second year of the study, the CURE survey was administered through Qualtrics. The means of the two years of data were combined using a weighted average and the standard deviations were pooled [40]. The data was analyzed using the two-sample *t*-test to analyze the differences in pre-course experience and post-course gains. *P*-values less than 0.05 were considered significant. A total of 38 students responded to the survey.

## 3. Results and Discussion

The curricular enhancements involve the integration of general chemistry II laboratory, organic chemistry II laboratory, and research writing courses in the form of a three-course learning community. These laboratory courses are populated by chemistry, biochemistry, physics, and engineering students and the course was run with a collaborative research project for two years. Prior to this time, the learning community was in place for three years, specifically integrating the first-year general education course, research writing, and the general chemistry II laboratory. While the organic chemistry laboratory was part of the learning community, there were no common assignments between all three courses. With the advent of adding research projects to the laboratory, the learning community became more robust, with common assignments and a common poster session in lieu of a final exam. During the progress of the semester, students met for common activities involving all three courses, such as a group writing workshop, a university-wide research poster session, and their final poster research session.

*3.1. Collaborative Research Project*

The overall goal of the research project was to utilize the potential of redox-active ligands to perform multi-electron processes in a similar way that transition metal catalysts perform chemical transformations. Transition metal complexes that perform small molecular transformations utilize multi-electron processes. Two-electron oxidations include C–H bond oxidation and the reduction of protons into dihydrogen, while the oxidation of water is a four-electron process and the reduction of dinitrogen is a six-electron process. To perform multi-electron transformations many times, one or more transition metals are implicated, invoking a change in the oxidation states of the metal or metals. Redox-active or non-innocent ligands containing oxygen, nitrogen, and sulfur donor atoms have gained considerable attention, recently, as ligands for transition metals, and may become involved in the electron transfer that is typically assigned to metal redox processes. This research project seeks to employ an alternative to a many-electron process involving transition metals by including redox-active ligands coordinated to the metal center to supply the necessary oxidative or reducing equivalents in order to perform desirable chemical transformations. Figure 1 shows the redox states of the chosen ligands for this study.

3.1.1. Role of the Organic Chemistry Laboratory:

The second semester organic chemistry laboratory focused on building skills in designing and carrying out organic synthesis projects. Students were assigned a project where their contributions were to develop and synthesize novel ligands that would be redox-active. Students used a chemical literature database to research redox-active ligands and learn how they are synthesized. The students selected a currently known redox-active ligand and then proposed a structural change that would

modify its activity. Catechols and aryl imines are known to be redox-active and most ligands chosen by students had this basic structure. A few of the target ligands had never been synthesized, which added to the excitement of the research experience.

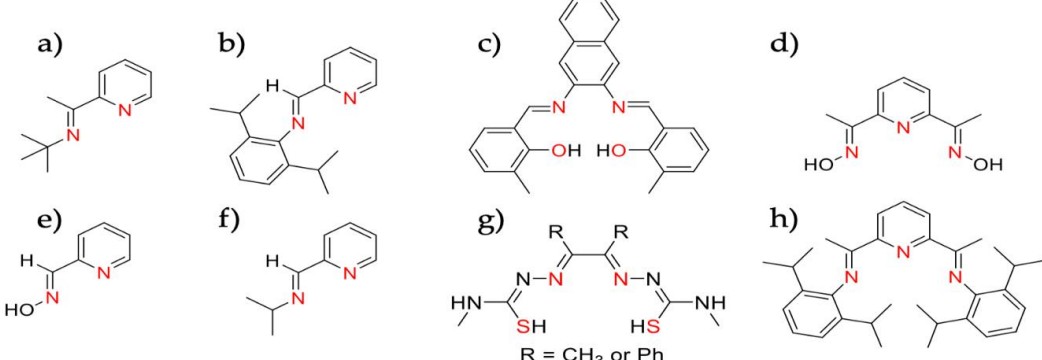

**Figure 1.** Schematic of redox states of N-based (**a**), O-based (**b**), and S-based (**c**) catecholate-type ligands.

The ligands that were synthesized by the organic chemistry students were nitrogen-, oxygen-, or sulfur-based catecholate or imine-based compounds. Figure 1 shows the redox-active nature of the catecholate-based ligands. Among this basic structure, there is a diverse array of possibilities, where R-groups and heteroatoms could be varied. Figure 2 shows some of the ligands that were synthesized by organic chemistry students.

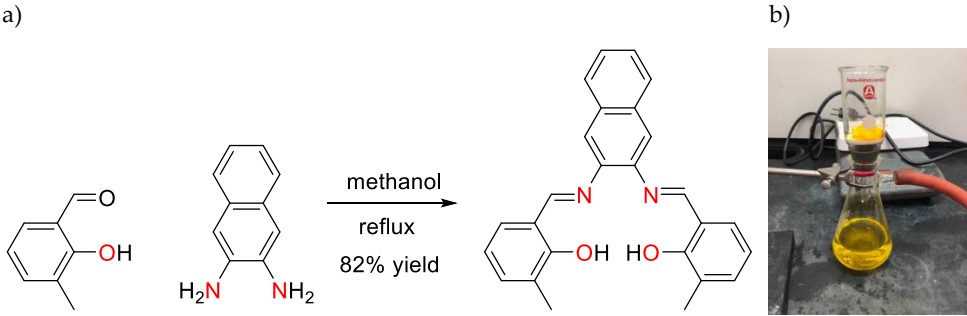

**Figure 2.** Representative imine-based ligands synthesized by second semester organic chemistry laboratory students.

A typical ligand synthesis is shown in Figure 3. Most syntheses required one or two reactions and were completed in three weeks. The ligands were purified using recrystallization or column chromatography.

a)                                                                                              b)

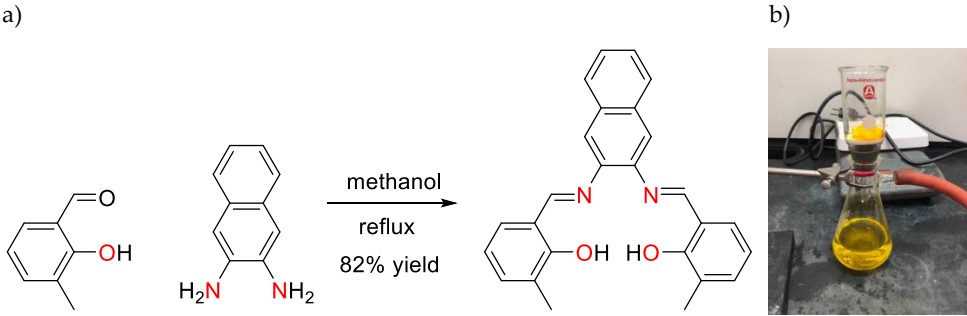

**Figure 3.** Scheme showing a typical synthesis of an imine based redox-active ligand (**a**) and filtration (**b**).

### 3.1.2. Role of the General Chemistry Laboratory

Once the organic chemistry students had synthesized their chosen ligand, purified and positively characterized the ligand structure using $^1$H Nuclear Magnetic Resonance spectroscopy, the general chemistry students proceeded by making metal–ligand complexes.

Prior to acquiring their ligands from the organic chemistry students, the general chemistry students were instructed on the basics of literature searching. Students formed groups of two and their first task was to formulate a research plan. They were given guidelines in which they would articulate their research plan based on the literature, outline their research activities for three weeks, and find five to 10 primary articles that supported their plan. The research groups were given parameters for the research project, in which they would make three choices for their research direction. The first choice that each group made was which ligand to focus their research experiments on. They could only choose ligands based on what the organic chemistry laboratory students did in the previous weeks. One of the goals of the research project was to investigate first-row transition metals; therefore, the next thing they had to do was choose three first-row transition metals to make metal complexes. Finally, they needed to decide on the reactivity in which they would investigate with their metal complexes. They could focus on oxygen, nitrogen, or hydrogen activation. Students were instructed to provide evidence based on the literature. The project had to strike a balance between the parameters of the research project, with three choices given to students to narrow the focus of their research investigations, while still giving students the freedom to explore their research questions. A typical synthesis of a transition metal complex with a redox-active ligand is shown in Figure 4.

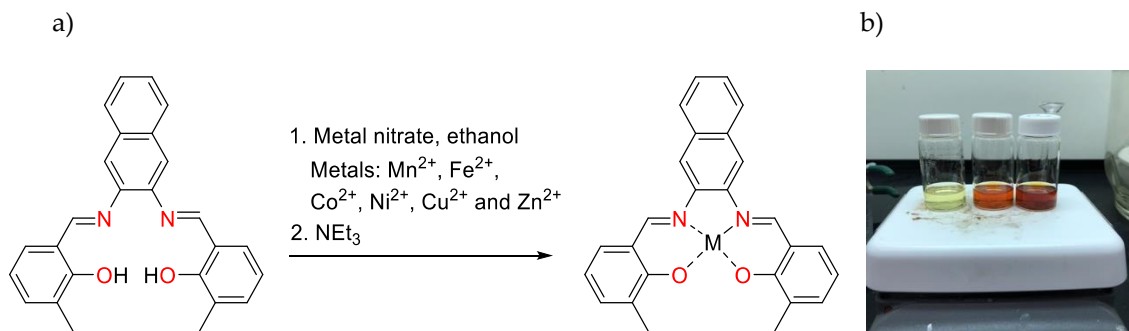

**Figure 4.** Scheme showing a typical synthesis of a metal–ligand complex using redox-active ligands and first-row transition metals (**a**) and, from left to right, $Zn^{2+}$, $Co^{2+}$, and $Fe^{2+}$ (**b**).

### 3.2. Layout and Timeline of the Learning Community

The three-course learning community with embedded authentic research experiences was piloted in the spring semester of 2018. Each lab session was three hours a week for 15 weeks. The timeline of the learning community activities and assignments is shown in Figure 5. During the course of the semester, each laboratory course had three research projects, but only one that was collaborative, which was the focus of this study. Students worked in pairs to create a more collaborative environment and also to promote the skill of working in groups. Students were assessed by the research proposal they formulated for each research project and at the end of the research project, through a journal-style paper in the format of the American Chemical Society. Each student practiced their scientific communication skills by giving oral presentations and a collaborative poster session that was held during their final exam time slot. The poster session was widely publicized to simulate a conference experience and was attended by the faculty and students in all the STEM disciplines.

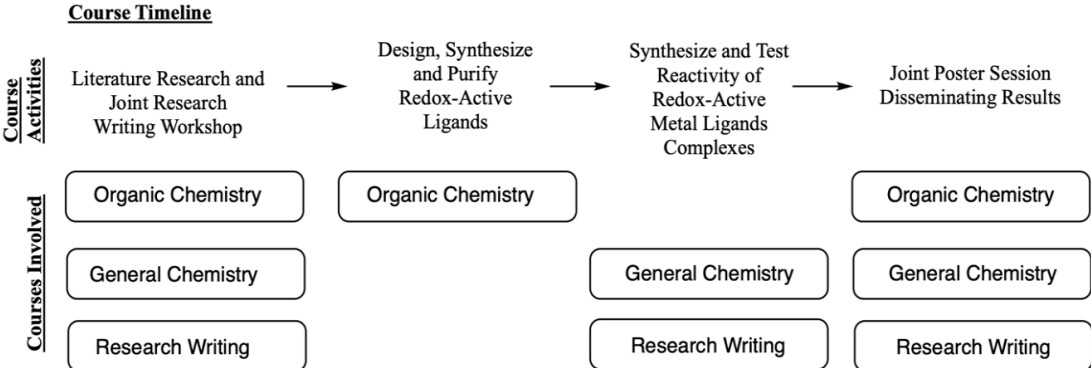

**Figure 5.** Timeline of the learning community between second semester general and organic chemistry laboratory and research writing courses.

Each of the course goals were assessed using the CURE survey. The overall goals of the course were as follows:

- Increased confidence in science process skills;
- Increased motivation to pursue further scientific research opportunities;
- The development of scientific communication skills.

The general chemistry laboratory II, organic chemistry laboratory II, and research writing courses were offered concurrently in the second semester of the academic year. There was a collaborative research project between the general and organic chemistry laboratories, where students worked on different aspects of the overall research project. Each laboratory had a total of three research modules throughout the semester; therefore, the organic chemistry students worked on the synthesis of redox-active ligands as their first project. After the organic students finished their part of the project, it was handed off to the general chemistry students, who then investigated the synthesis of first-row transition metal complexes and reactivity with respect to oxygen, hydrogen, or nitrogen activation. There were common assignments linking the three courses throughout the semester, as shown in Figure 6.

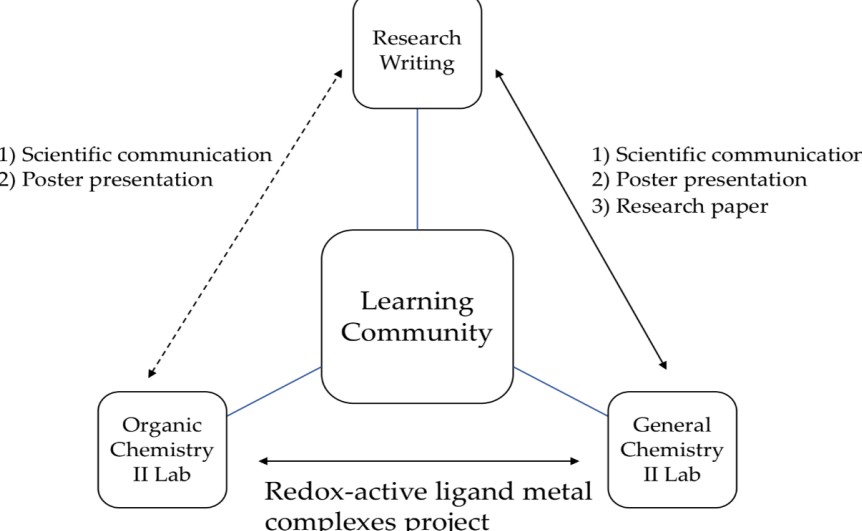

**Figure 6.** Three-course learning community scheme, indicating linked activities and assignments. Solid lines indicate direct relationships, where a common assignment is turned in to each class, while dashed lines indicate indirect relationships between courses, which means the experience was shared, but there was not a shared assignment between courses.

### 3.2.1. Course Goal: Increased Confidence in Science Process Skills.

The course activities and assessments in the three-course learning community and in individual courses aimed to increase confidence in scientific process skills. Course elements related to scientific process skills, based on the elements of a CURE, are as follows: (1) writing a research proposal, (2) analyzing data, (3) reading primary literature, and (4) working in small groups. Table 1 shows a comparison between the CURE surveys taken by the general chemistry laboratory students before and after the course, and Table 2 shows the CURE survey data results for the organic chemistry laboratory students before and after the course. The assignments shared by the learning community involved writing a journal-style research paper and a poster presentation. All students worked in groups of two or three; therefore, working in small groups was emphasized throughout the entire semester. All of these elements presented statistically significant post-course gains relative to pre-course experiences. Most importantly, the students learned how to formulate a research plan and write a research proposal based on reading primary scientific literature. These skills saw statistically significant gains, as judged by the percentage increase from the pre-course experience and the post-course gains, indicating that the students had minimal prior exposure to these types of assignments.

**Table 1.** CURE survey data of second semester general chemistry laboratory students for 2018 and 2019.

| 5 Elements of a CURE | | Course Elements | Pre-Course Mean | Post-Course Mean | Percent Increase | *p*-Value |
|---|---|---|---|---|---|---|
| Scientific practices | Forming hypotheses | Write a research proposal | 2.15 | 4.10 | 91% | <0.001 |
| | Collecting and analyzing | Analyze data | 3.89 | 4.68 | 20% | 0.001 |
| | Communicating findings | Present posters | 2.33 | 4.61 | 98% | <0.001 |
| Discovery of new knowledge | | At least one project assigned and structured by instructor | 3.68 | 4.10 | 12% | 0.115 [1] |
| Research that is broadly relevant | | Reading primary scientific literature | 2.65 | 4.68 | 77% | <0.001 |
| Collaboration between students | | Work in small groups | 3.89 | 4.78 | 23% | 0.001 |
| Iterative process | | Critique work of other students | 2.98 | 3.41 | 15% | 0.191 [1] |

[1] *p*-values greater than 0.05 are considered to be insignificant.

**Table 2.** Course-Based Undergraduate Research Experience (CURE) survey data of second semester organic chemistry laboratory students for 2018 and 2019.

| 5 Elements of a CURE | | Course Elements | Pre-Course Mean | Post-Course Mean | Percent Increase | *p*-Value |
|---|---|---|---|---|---|---|
| Scientific practices | Forming hypotheses | Write a research proposal | 3.02 | 4.56 | 51% | <0.001 |
| | Collecting and analyzing | Analyze data | 3.90 | 4.82 | 24% | 0.002 |
| | Communicating findings | Present posters | 2.65 | 4.70 | 78% | <0.001 |
| Discovery of new knowledge | | At least one project assigned and structured by instructor | 3.91 | 4.24 | 8% | 0.379 [1] |
| Research that is broadly relevant | | Reading primary scientific literature | 3.81 | 4.71 | 23% | 0.005 |
| Collaboration between students | | Work in small groups | 4.17 | 4.77 | 14% | 0.012 |
| Iterative process | | Critique work of other students | 2.59 | 3.13 | 21% | 0.282 [1] |

[1] *p*-values greater than 0.05 are considered to be insignificant.

### 3.2.2. Course Goal: Increased Motivation to Pursue Further Scientific Research Opportunities.

Student motivation to pursue further scientific research opportunities was gauged by their application and participation in summer research opportunities. Benedictine University has a strong

culture in engaging students in undergraduate research. The Benedictine University Natural Sciences Summer Research Program (NSSRP) is a longstanding program that financially supports ~25 students each summer to participate in a 10-week summer research program with faculty mentors. Students are actively recruited by the faculty to participate in this program. In 2016 and 2017, no students from the section of organic chemistry comprised of chemistry and biochemistry majors applied to the summer research program. After the implementation of the collaborative Course-Based Research Experience learning community, there was a drastic increase in student motivation to pursue further scientific research opportunities. In 2018, 64% of students applied to the NSSRP and 57% participated. In 2019, 44% of organic chemistry students applied to the NSSRP and 31% participated in the program.

### 3.2.3. Course Goal: Development of Scientific Communication Skills

Another linked assignment in the three-course learning community is a joint poster session at the end of the semester. Both the general and organic chemistry laboratory students present posters of their research projects in place of a final examination for the course. The faculty from the entire college are invited to view the posters and ask students questions. This allowed the community to really come together, with a feeling of unification. Organic students recognized the ligands they had synthesized on the posters presented by the general chemistry students' metal–ligand complexes. Course elements related to scientific communication skills based on the elements of a CURE are as follows: (1) presenting posters and (2) the critique of other students. Again, drawing from the CURE survey results shown in Tables 1 and 2, only the skill of presenting posters saw a large gain between pre-course experience and post-course gain that was statistically significant. This result is not surprising, since the course activities mandated two poster presentations during the course of the semester.

### 3.3. Overall Assessment of the 3-Course Learning Community

The CURE survey was developed by David Lopatto and was administered by his team until 2018. After that time, the survey could be used on our own survey server and analyzed by our team. The CURE survey was administered in spring 2018 and 2019 to general and organic chemistry laboratory students. These courses were populated by chemistry, biochemistry, physics and engineering majors, with a total of 38 students participating.

The goals of the three-course learning community align with the skill sets gained from a research or research-like experience. Each of the goals were evaluated by either results obtained from the CURE survey or by the increase in the number of students pursuing internal or external research experiences. The data shows statistically significant pre-course experiences to post-course gains in five of the areas that were aligned with the five elements of a CURE: (1) write a research proposal, (2) analyze data, (3) present posters, (4) reading primary scientific literature, and (5) work in small groups. All of these course elements were emphasized throughout the semester and gains were large in some cases. This indicates that, prior to their laboratory experience, the students had little exposure to the scientific processes and practices performed by practitioners in their fields.

### Student Survey Feedback

The benefits of involving students in research experiences are well known. These benefits include increased self-efficacy, self-identification as a scientist, and confidence in scientific process and practices [6,7,9,27,36–39,41]. In addition to the CURE survey, students were asked to respond to survey prompts inquiring about their impressions of their research experiences and how the experience may have impacted their future goals. In the survey, students were asked how they felt about the course and were offered an opportunity to provide open-ended responses to survey prompts. There were some interesting insights that came out of this qualitative analysis.

- "The research experience influenced me by demonstrating the meaningful work that can be done in my future career."

- "This was a great experience and I would love to do it again."
- "I love research!"
- "This confirmed my love for science and chemistry and therefore increased my interest in applying to pharmacy school."
- "I had fun being a part of the "research" experience."
- "I would definitely recommend it to other students!"
- "My self-confidence increased!"
- "I learned a lot about research and I had a lot of fun with all of the projects."

These comments from the students led us to look into students' interest in research experiences. As mentioned in Section 3.2.2, an uptick in interest in pursuing additional research experiences was observed for students that participated in research projects in their course laboratory work.

## 4. Conclusions

Authentic research experiences and learning communities are high-impact educational practices that have been shown to result in the higher retention of students that declare majors in STEM disciplines. A collaborative course-based research experience for general and organic chemistry laboratory students, who are involved in a three-course learning community within their research writing course, is described in this paper. Common assignments in the learning community include writing group research reports and poster presentations. Based on these programmatic enhancements of incorporating a learning community and adding authentic and collaborative research experiences, the data support our prior assumptions that these experiential enhancements increase confidence in science process skills and support students' interest in pursuing additional research experiences.

According to the persistence framework, the retention of students in STEM disciplines is related to learning and professional identification, where both of these increase confidences in scientific process and practices and motivation towards further learning and opportunities. This study shows that courses that use authentic research projects that are linked within a three-course learning community lead to experiential gains for students, which are aligned to the five CURE elements and an increased interest in pursuing additional research experiences. While direct measurements of retention require a longer-term study, our data indicated that the key factors leading to the increased retention of students in STEM disciplines are improved.

**Author Contributions:** Research conceptualization, K.L.S. and D.M.R.; methodology, K.L.S.; formal analysis, K.L.S.; investigation, K.L.S. and D.M.R.; writing—original draft preparation, K.L.S.; writing—review and editing, K.L.S. and D.M.R. All authors have read and agreed to the published version of the manuscript.

**Funding:** The APC was funded by Lewis University.

**Acknowledgments:** The authors would like to acknowledge David Lopatto and Leslie Jaworski for data analysis of the CURE survey and David Stone for statistical analysis of the CURE survey data.

**Conflicts of Interest:** The authors declare no conflict of interest.

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
