# Peer review of "A Learning Community Involving Collaborative Course-Based Research Experiences for Foundational Chemistry Laboratories"

_education, doi:10.3390/educsci10040117_

Round 1

Reviewer 1 Report

Review of EDUCATION-772255

This study describes the use of course embedded undergraduate research and learning communities in three courses including second term organic and general chemistry.  Students in organic chemistry designed and synthesized ligands which were used in a second module by general chemistry students to develop redox active metal-ligand complexes.  The idea behind this work is original and impressive, and there is much to like about it.  I have some suggestions to improve the manuscript:

  • One of the primary outcomes of this work was an uptick in students from the organic course applying to the university summer research program. However, it seems difficult to prove that this is due to the project directly, or at least no evidence was presented to say that this was so.  Perhaps students were encouraged to apply, whereas they had not been previously.  Can the authors present additional evidence or comment on this section?

  • The CURE survey results seem robust, but regarding student evaluations, were there numeric data collected as well from a standardized set of questions? The comments are helpful but may not be entirely representative.

  • As the authors, note, the longer term benefits of this work center on retention within the majors but a side benefit, as the authors indicate, is student participation in research, which is a known high impact practice. Perhaps the authors could comment further on this, and more generally on the culture of research at their institutions.

  • For those interested in implementing CURE in gen chem and organic laboratories, It would be appreciated to hear more detail about the projects in the individual courses. For example, how did organic students select their ligands and the synthetic approach?  How did this tie into the learning objectives of that course? 

Author Response

We appreciate your constructive feedback of our manuscript. Your comments are bolded and our responses are given in black text with no highlight. Noted below is how we have revised the manuscript to reflect your comments as follows:

This study describes the use of course embedded undergraduate research and learning communities in three courses including second term organic and general chemistry.  Students in organic chemistry designed and synthesized ligands which were used in a second module by general chemistry students to develop redox active metal-ligand complexes.  The idea behind this work is original and impressive, and there is much to like about it.  I have some suggestions to improve the manuscript:

  • One of the primary outcomes of this work was an uptick in students from the organic course applying to the university summer research program. However, it seems difficult to prove that this is due to the project directly, or at least no evidence was presented to say that this was so.Perhaps students were encouraged to apply, whereas they had not been previously.  Can the authors present additional evidence or comment on this section?

Added to Lines 227-231: additional details about research culture and recruiting students to the summer research program.

  • The CURE survey results seem robust, but regarding student evaluations, were there numeric data collected as well from a standardized set of questions? The comments are helpful but may not be entirely representative.

See table 2 for numeric data. In addition, the manuscript was edited in Lines 273-276 to provide more clarity on the kinds of survey prompts.

  • As the authors, note, the longer term benefits of this work center on retention within the majors but a side benefit, as the authors indicate, is student participation in research, which is a known high impact practice. Perhaps the authors could comment further on this, and more generally on the culture of research at their institutions.

Added to Lines 227-231: additional details about research culture and recruiting students to the summer research program.

  • For those interested in implementing CURE in gen chem and organic laboratories, It would be appreciated to hear more detail about the projects in the individual courses. For example, how did organic students select their ligands and the synthetic approach?How did this tie into the learning objectives of that course?

Lines: 125-130: Added additional clarification on the projects in the organic chemistry laboratory as well as how these tie into the learning objectives for the course.

Reviewer 2 Report

The introduction does a good job of introducing the rational and motivation for implementing a learning community and CURE in foundational chemistry labs, but does not specifically discuss any other attempts to create CUREs in introductory chemistry courses.  I was also a bit confused about what exactly constitutes a "learning community" as it appears that the students in the two chemistry courses do not interact directly at all until the final poster session and the authors never described or linked the writing course to these two chemistry courses.  I was left assuming that the writing course was taken simultaneously by the General Chemistry II lab students.  The authors need to include a description of the writing course and how it is integrated with the two chemistry courses.  Section 3.2.3 mentions that the course activities included "two poster presentations" and I only recall description of the final poster session.  So what is the 2nd poster presentation?

Minor comments:

Figure legend for Fig. 4 is missing a description for part (b).

Line 207, pg 6 - remove "and" after "Course"

Author Response

We appreciate your constructive feedback of our manuscript. Your comments are bolded and our responses are given in black text with no highlight. Noted below is how we have revised the manuscript to reflect your comments as follows:

The introduction does a good job of introducing the rational and motivation for implementing a learning community and CURE in foundational chemistry labs, but does not specifically discuss any other attempts to create CUREs in introductory chemistry courses.  I was also a bit confused about what exactly constitutes a "learning community" as it appears that the students in the two chemistry courses do not interact directly at all until the final poster session and the authors never described or linked the writing course to these two chemistry courses.  I was left assuming that the writing course was taken simultaneously by the General Chemistry II lab students.  The authors need to include a description of the writing course and how it is integrated with the two chemistry courses.  Section 3.2.3 mentions that the course activities included "two poster presentations" and I only recall description of the final poster session.  So what is the 2nd poster presentation?

Lines 100-110: Additional information as to how all three courses are connected to each other.

Minor comments:

Figure legend for Fig. 4 is missing a description for part (b).

Fixed.

Line 207, pg 6 - remove "and" after "Course"

Fixed.

Reviewer 3 Report

This paper is centred in a very relevant topic for those involved in curricular development in Higher Education Science courses: The integration of a learning community and authentic research experiences to laboratory courses in order to promote greater post-course gains of essential elements of a CUREs curriculum and higher retention of students in STEM courses.

The paper describes and evaluates the development of a course. However, in my opinion, the article lacks a clear definition of the research questions/intentions, especially in what concerns the evaluation component.

The review of the literature is adequate and relevant to the project description and evaluation.

Both the data collection and the data analysis procedures are adequate to the research.

The results and the conclusions are interesting and relevant for those involved in the implementation of similar courses in universities.

Author Response

We appreciate your constructive feedback of our manuscript. Your comments are bolded and our responses are given in black text with no highlight. Noted below is how we have revised the manuscript to reflect your comments as follows:

This paper is centred in a very relevant topic for those involved in curricular development in Higher Education Science courses: The integration of a learning community and authentic research experiences to laboratory courses in order to promote greater post-course gains of essential elements of a CUREs curriculum and higher retention of students in STEM courses.

The paper describes and evaluates the development of a course. However, in my opinion, the article lacks a clear definition of the research questions/intentions, especially in what concerns the evaluation component.

Lines 260-262: Provides clarification of the 3-course learning community evaluation process.

The review of the literature is adequate and relevant to the project description and evaluation.

Both the data collection and the data analysis procedures are adequate to the research.

The results and the conclusions are interesting and relevant for those involved in the implementation of similar courses in universities.